# An Application of Reinforced Learning-Based Dynamic Pricing for Improvement of Ridesharing Platform Service in Seoul

**Jaein Song** [1] , **Yun Ji Cho** [2], **Min Hee Kang** [2] and **Kee Yeon Hwang** [3,*]

1. Research Institute of Science and Technology, Hongik University, Seoul 04066, Korea; wodlsthd@nate.com
2. Department of Smart City, Hongik University, Seoul 04066, Korea; yunji9943@gmail.com (Y.J.C.); speakingbee@hanmail.net (M.H.K.)
3. Department of Urban Planning, Hongik University, Seoul 04066, Korea
* Correspondence: keith@hongik.ac.kr

**Abstract:** As ridesharing services (including taxi) are often run by private companies, profitability is the top priority in operation. This leads to an increase in the driver's refusal to take passengers to areas with low demand where they will have difficulties finding subsequent passengers, causing problems such as an extended waiting time when hailing a vehicle for passengers bound for these regions. The study used Seoul's taxi data to find appropriate surge rates of ridesharing services between 10:00 p.m. and 4:00 a.m. by region using a reinforcement learning algorithm to resolve this problem during the worst time period. In reinforcement learning, the outcome of centrality analysis was applied as a weight affecting drivers' destination choice probability. Furthermore, the reward function used in the learning was adjusted according to whether the passenger waiting time value was applied or not. The profit was used for reward value. By using a negative reward for the passenger waiting time, the study was able to identify a more appropriate surge level. Across the region, the surge averaged a value of 1.6. To be more specific, those located on the outskirts of the city and in residential areas showed a higher surge, while central areas had a lower surge. Due to this different surge, a driver's refusal to take passengers can be lessened and the passenger waiting time can be shortened. The supply of ridesharing services in low-demand regions can be increased by as much as 7.5%, allowing regional equity problems related to ridesharing services in Seoul to be reduced to a greater extent.

**Keywords:** dynamic pricing; reinforcement learning; ridesharing; supply improvement; taxi

## 1. Introduction

Since the 1960s, with the beginning of industrialization, Korea experienced a rapid urbanization [1–3]. As such, the transportation infrastructure has also developed steadily, greatly improving the mobility of citizens. However, in metropolitan areas, such as the capital city and its surrounding regions, the expansion of urban areas and overpopulation caused increased traffic and consequently other social problems, including traffic congestion and air pollution. To minimize such social costs, various policies to facilitate public transportation and taxis have been actively promoted [4]. Despite the efforts, the current services are short in providing an equal level of mobility to all residents, requiring enhancement. In particular, in Seoul Metropolitan City, despite its excellent public transportation infrastructure, there is a constant inconvenience of users with a need for enhancing problems caused by congestion during rush hours and a shortage of ridesharing (including taxi) supply late at night on the outskirts of the city and in hilly areas [5].

Against this backdrop and with the recent introduction of the mobility as a service(MaaS) concept, efforts have been made to improve individuals' mobility and accessibility through the integrated

use of shared and cutting-edge modes of transportation [6–8]. MaaS consists of various means of transportation, including public transportation and a mobile platform-based ridesharing service. However, the ridesharing service is usually owned by the private sector, oriented toward profitability. Therefore, the provision of such a service tends to be concentrated in areas with high demand. Even if the ridesharing service were to expand, those users living in areas with low demand, especially suburban areas, are expected to face inconvenient services, such as longer waiting times due to the lower availability of vehicles. Especially during late-night hours when public transportation no longer operates, there are only limited options available, thus leading to further exacerbation of the imbalance between supply and demand. Therefore, a solution for this problem is needed. In a previous study [9] on experiences with taxi service refusal, respondents answered that they had difficulties getting taxi services or were refused during late-night hours (48.7%) when their destinations were either remote areas or outside of the city boundary (32.8%). From the perspective of users, this is recognized as a serious problem. The purpose of this study was to develop a dynamic pricing scheme to attract ridesharing (including taxi) drivers to mobility-disadvantaged regions during late-night hours when public transportation services are either reduced or come to a halt.

In the meantime, with the recent Fourth Industrial Revolution, the overall social paradigm has changed. A hyperintelligent society based on cutting-edge technologies, such as artificial intelligence, is being implemented. In the transportation sector, studies related to the introduction of big data- and artificial intelligence-based technology are growing [10]. Indeed, overseas ridesharing platforms, namely, Uber and Lyft, optimize supply and demand, as well as profitability, through AI-based dynamic pricing to retain the number of vehicles needed to respond to high demand in certain regions. Among the various subcategories of artificial intelligence, reinforcement learning has advantages in exploring unknown areas and identifying optimal outcomes through repeated exploration, unlike supervised learning, which is based on already available data. In contrast to other countries, there are not enough data related to ridesharing services in Korea and, accordingly, reinforcement learning would be more suitable. Consequently, this study conducted a reinforcement learning method using Seoul's taxi data to determine regionally appropriate levels of dynamic ridesharing fare rates with the purpose of improving the supply of ridesharing services in the mobility-disadvantaged regions in Seoul.

The paper is organized as follows: Section 2 reviews studies on the spatial fairness issues of transportation services and dynamic pricing applications for ridesharing services. Section 3 discusses the analysis framework and reinforcement learning method applied in this study. Subsequently, the results of the analysis are presented and discussed in Section 4, while the conclusions are drawn in Section 5.

## 2. Literature Review

### 2.1. Studies Analyzing Spatially Marginalized Areas in Terms of Transportation Services

In their study, Lee et al. [11] identified the mobility of different marginalized groups using smart card data and evaluated the mobility of groups highly reliant on public transportation. Building upon this, the study also categorized regions into different types according to their need for mobility improvement and concluded that mainly the outskirt areas of a city require improvements. Ha et al. [12] located areas marginalized from public transportation services in Seoul, using real travel characteristics gathered using Google and T map navigation application programming interfaces (APIs), and they analyzed areas with high priority in service enhancement. Data, such as travel characteristics and socioeconomic indices, were analyzed, identifying Gangbuk-gu, Seongbuk-gu, Seodaemun-gu, Jungnang-gu, and the southeast zone as areas requiring improvements in public transportation. Han [13] deduced spatially marginalized areas by assessing user mobility and the service level of providers. Moreover, the study reviewed the potential equity issue caused among different groups and suggested ways to improve it. According to the study, overall public transportation showed a satisfactory level from the users' perspective; however, there was a large deviation in terms of the

supply level by region. Notably, on the outskirts of the city, including Nowon-gu and Gwanak-gu, the gap between supply and demand appeared to be wide and, thus, most urgently requiring an improvement in supply.

From an accessibility perspective, Lee et al. [14] developed various indices for the connectivity, directness, and diversity of public transportation using transportation card data and assessed each transportation zone. This study confirmed that, with a higher connectivity of public transportation, it had more routes and better directness. On the contrary, in zones located on the outskirts of the city, marginalization from public transportation was vivid. Kim et al. [15] and Yoon et al. [16] studied regional marginalization and inequity by considering not just spatial accessibility, but also social classes. Kim et al. overlaid and compared socioeconomic characteristics and city zones using location data of public transportation in Daegu. According to their analysis, the low accessibility and environmental inequity of socially disadvantaged people (the aged, recipients of national basic livelihood benefits, etc.) in suburban areas were confirmed. Yoon et al. calculated the inequity index of socially marginalized people on the basis of a Gini-style index and the methods of accessibility measure developed by Curie. As for public transportation accessibility, the regional gap was bigger for the subway than the bus. When this was overlaid on top of the data for the socially disadvantaged group, inequity was confirmed to be greater for the subway than the bus.

There were studies assessing equity depending on regional differences in infrastructure. Kim et al. [17] analyzed disadvantaged regions by overlaying service areas of public transportation and confirmed that suburb areas were mainly in a disadvantageous position. Furthermore, when considering socioeconomic characteristics, the study concluded that there was a gap in public transportation infrastructure among regions. Lee et al. [18] used Seoul Metropolitan Household Travel Survey data to measure regional equity among different income brackets depending on the levels of transportation infrastructure. The study showed that lower spatial equity led to longer total traveling time. Bin et al. [19] carried out a spatial cluster analysis at the administrative unit (Eup, Myeon, and Dong) level with transportation infrastructure indicators and travel behavior to assess equity in Gyeonggi-do province (excluding Seoul Metropolitan City and Incheon City). The results clearly showed gaps between areas closer to Seoul and those on the outskirts of the city. In particular, equity at the infrastructure level in northern Gyeonggi-do province was low.

## 2.2. Dynamic Pricing Studies on Ridesharing Service

Before reviewing previous literature, dynamic pricing can be defined as a strategy in which prices change flexibly for the same product or service depending on the market situation [20–22]. This strategy is mainly employed with respect to e-commerce, flight tickets, and hotel booking and demand management. This results in the optimization of selling products and services in an environment where the price can be easily adjusted. With regard to dynamic pricing for ridesharing services, there were studies on profitability improvement and the determination of an appropriate price through a pricing strategy [23–30], studies analyzing the elements of a pricing strategy affecting customers or drivers [31–33], and a study based on reinforcement learning [34].

First, Banerjee et al. [23] validated the performance of dynamic pricing by suggesting a ridesharing model considering two aspects: the stochastic dynamics of the market and the strategic decisions of the drivers, passengers, and platform. According to the analysis, depending on supply and demand conditions, flexible pricing resulted in increased total utility. Zeng et al. [24] researched the dynamic pricing strategy in accordance with potential users considering the destination of taxies. Markov Decision Process (MDP) was established by considering the cost of pick-ups at the destination. The total utility was enhanced compared to the fixed cost case. Hall et al. [25] studied the economic utility of surge pricing by analyzing Uber data.

When prices rise in line with the surge price algorithm, the delta affected the improving profitability of drivers, supply, and efficiency. Moreover, if surge pricing was not employed during

peak hours, passenger waiting time was extended as drivers did not pick up passengers and passenger utility dropped.

In an analysis of the effects of dynamic pricing on drivers through Uber cases, Chen et al. identified that the surge price has a negative impact on passengers and a positive impact on drivers. Furthermore, the study found unfairness along the regional border of surge pricing. Chen et al. [32] studied changes in the number of Uber drivers depending on changes in surge price. The analysis confirmed that there was a higher rate of vehicle operation, as well as changes in operating hours, during the time period in which higher profit was expected from the drivers' perspective. Kooti et al. [33] analyzed the impact of dynamic pricing and income on the behavior of drivers, and they found that drivers operating vehicles during peak hours earned a higher income compared to nonpeak-hour drivers.

Wu et al. [34] simulated the application of dynamic pricing to ridesharing services. They compared four pricing methodologies, namely, (1) statistic pricing, (2) proportional pricing, (3) batch updates, and (4) reinforcement learning with the goal of profit maximization in a single Origin-Destination(OD) scenario. The simulation found that pricing based on reinforcement learning increased the total profit and individual profit of drivers the most.

### 2.3. Summary of the Review

Previous studies on mobility-disadvantaged regions mainly focused on evaluating areas marginalized from public transportation from a user's perspective utilizing big data, such as smart card data, household travel surveys, and Geographic Information System Data Base (GIS DB: location data for bus and subway, thematic transportation map, etc.). These studies identified that public transportation services in these regions were not adequately provided. On the other hand, they paid little attention to the late-night mobility inconvenience in regions after public transportation services had ended.

As for dynamic pricing in ridesharing services, studies were carried out mainly using Uber cases. As Uber discloses operation data, several studies related to dynamic pricing were conducted. In particular, the surge-based pricing analysis of Uber showed that it helped increase driver supply and operational profit. On the contrary, if a single-fare system was applied without flexibility, overall utility decreased due to a lower rate of matching and a longer waiting time when suppliers decided not to take passengers. However, there were no studies measuring regional differences in dynamic pricing; thus, it is necessary to determine ways of addressing drivers' refusal to take passengers when taking regional differences into account.

Accordingly, a study evaluating the dynamic pricing solution is required to improve the quality of mobility services in disadvantaged regions in Seoul.

## 3. Analysis Methodology

To simulate dynamic pricing using reinforcement learning, this study first conducted a centrality (in-bound and out-bound) analysis with a district(gu)-level OD matrix to develop a regional indicator of degree centrality to be used in reinforcement learning analysis. Then, the indicators were applied to a reinforcement learning simulation.

### 3.1. Analysis Methodology

#### 3.1.1. Degree Centrality Analysis Methodology

Degree centrality is an indicator showing how many nodes are connected to a certain node. In social network theory, degree centrality is defined as the number of nodes linked directly to any given node. This study measured the out-degree and in-degree of each region (node) from a centrality

perspective depending on the level of node connectivity, using them as indicators to be applied in reinforcement learning. Centrality can be calculated as follows:

$$C'_D(N_i) = \frac{\sum_{j=1}^{g} x_{ij}}{g-1}, \; i \neq j, \tag{1}$$

where $C'_D$ is the standardized degree centrality of node $i$, $\sum_{j=1}^{g} x_{ij}$ is the degree centrality of node $j$, and $g$ is the number of nodes. If there is no direction in the network, the above equation simply shows the degree of nodes. On the other hand, if directions are present, the equation distinguishes out-degree centrality ($C'_{outD}$) and in-degree centrality ($C'_{inD}$). Here, out-degree centrality is defined as the level of connections going out from a certain node to other nodes, and in-degree centrality is defined as the level of connections coming in from other nodes to a certain node. In this study, in-degree centrality referred to the number of vehicles coming into a certain zone (Gu), thus indicating a region as a frequently selected destination by passengers (usually residential areas during late-night periods). On the contrary, out-degree centrality referred to the number of vehicles traveling out of a certain zone, thus indicating a region as a preferred destination by drivers (central and subcentral regions). Therefore, this study identified indicators by region considering both out-degree and in-degree centrality.

### 3.1.2. Reinforcement Learning Methodology

Reinforcement learning is a learning method through which an agent chooses an action to take in an environment to maximize reward. This affects not only the immediate reward due to the action of the agent, but also the long-term reward. The main characteristics of reinforcement learning are a trial-and-error search and delayed reward (Figure 1). This study used the Q-learning algorithm for analysis.

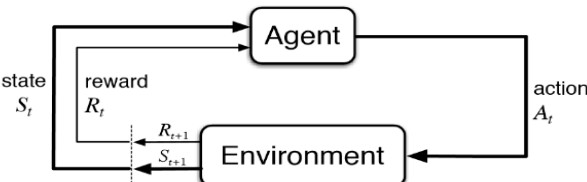

**Figure 1.** Conceptual diagram of reinforcement learning.

In the Q-learning algorithm [35,36], Q initially has an arbitrary fixed value. When the agent selects an action ($a$) in accordance with a learning step ($t$), an immediate reward ($r$) is observed before entering a new state ($s$), updating the Q value. The key characteristic of this algorithm is the value iteration update, using the weighted average of the old value and new information. The equation of Q-learning can be expressed as follows:

$$Q(s_t, \, a_t) \leftarrow (1-\alpha)Q(s_t, \, a_t) + \alpha(r_t + \gamma \max_{\alpha} Q(s_{t+1}, \, a), \tag{2}$$

where $\alpha$ is the learning rate which updates the current Q value with immediate reward and/or future expected value, usually between 0 and 1. When the value is 0, the current Q value is continuously used without any update according to learning. On the contrary, if the value is 1, the previous Q value is ignored and updated automatically. $\gamma$ is a discount factor which is a variable explaining the difference between immediate and future rewards, also between 0 and 1. When the value is 0, the agent takes a myopic action as by making an update with immediate reward. When the value is 1, the agent leans toward a future reward, underestimating the current reward. In this study, on the basis of existing studies, we experimentally set the learning rate to 0.1 and discount factor to 0.9.

### 3.2. Environment Setting for Reinforcement Learning Simulation

3.2.1. Q-Table Configuration

To carry out the reinforcement learning simulation using the Q-learning algorithm, a Q-table needed to be defined through state and action. The Q-table is a matrix that compiles rewards for all states and actions, and it is updated after each learning step. Here, 600 OD cells were used in an OD matrix of 25 zones in Seoul excluding intrazonal travel. The state refers to the price that can potentially be generated in each OD cell depending on time slots, which can be expressed as 13 (price range) × 6 h (time round, Tr) for each OD cell. In total, there were 13 actions, including a standard price and surge prices from 0.6-fold to 3.0-fold the standard price with an interval of 0.2. The composition of the district OD matrix is shown in Figure 2, with a separate layer constructed for each cell's travel time and passage cost ($P_{base}$) to be used for analysis.

|  | Z1 | Z2 | Z3 | Z4 | Z5 | Z6 | Z7 | Z8 | Z9 | Z10 | Z11 | Z12 | Z13 | Z14 | Z15 | Z16 | Z17 | Z18 | Z19 | Z20 | Z21 | Z22 | Z23 | Z24 | Z25 |
|---|---|---|---|---|---|---|---|---|---|---|---|---|---|---|---|---|---|---|---|---|---|---|---|---|---|
| Z1 |  | 0 | 1 | 2 | 3 | 4 | 5 | 6 | 7 | 8 | 9 | 10 | 11 | 12 | 13 | 14 | 15 | 16 | 17 | 18 | 19 | 20 | 21 | 22 | 23 |
| Z2 | 24 |  | 25 | 26 | 27 | 28 | 29 | 30 | 31 | 32 | 33 | 34 | 35 | 36 | 37 | 38 | 39 | 40 | 41 | 42 | 43 | 44 | 45 | 46 | 47 |
| Z3 | 48 | 49 |  | 50 | 51 | 52 | 53 | 54 | 55 | 56 | 57 | 58 | 59 | 60 | 61 | 62 | 63 | 64 | 65 | 66 | 67 | 68 | 69 | 70 | 71 |
| Z4 | 72 | 73 | 74 |  | 75 | 76 | 77 | 78 | 79 | 80 | 81 | 82 | 83 | 84 | 85 | 86 | 87 | 88 | 89 | 90 | 91 | 92 | 93 | 94 | 95 |
| Z5 | 96 | 97 | 98 | 99 |  | 100 | 101 | 102 | 103 | 104 | 105 | 106 | 107 | 108 | 109 | 110 | 111 | 112 | 113 | 114 | 115 | 116 | 117 | 118 | 119 |
| Z6 | 120 | 121 | 122 | 123 | 124 |  | 125 | 126 | 127 | 128 | 129 | 130 | 131 | 132 | 133 | 134 | 135 | 136 | 137 | 138 | 139 | 140 | 141 | 142 | 143 |
| Z7 | 144 | 145 | 146 | 147 | 148 | 149 |  | 150 | 151 | 152 | 153 | 154 | 155 | 156 | 157 | 158 | 159 | 160 | 161 | 162 | 163 | 164 | 165 | 166 | 167 |
| Z8 | 168 | 169 | 170 | 171 | 172 | 173 | 174 |  | 175 | 176 | 177 | 178 | 179 | 180 | 181 | 182 | 183 | 184 | 185 | 186 | 187 | 188 | 189 | 190 | 191 |
| Z9 | 192 | 193 | 194 | 195 | 196 | 197 | 198 | 199 |  | 200 | 201 | 202 | 203 | 204 | 205 | 206 | 207 | 208 | 209 | 210 | 211 | 212 | 213 | 214 | 215 |
| Z10 | 216 | 217 | 218 | 219 | 220 | 221 | 222 | 223 | 224 |  | 225 | 226 | 227 | 228 | 229 | 230 | 231 | 232 | 233 | 234 | 235 | 236 | 237 | 238 | 239 |
| Z11 | 240 | 241 | 242 | 243 | 244 | 245 | 246 | 247 | 248 | 249 |  | 250 | 251 | 252 | 253 | 254 | 255 | 256 | 257 | 258 | 259 | 260 | 261 | 262 | 263 |
| Z12 | 264 | 265 | 266 | 267 | 268 | 269 | 270 | 271 | 272 | 273 | 274 |  | 275 | 276 | 277 | 278 | 279 | 280 | 281 | 282 | 283 | 284 | 285 | 286 | 287 |
| Z13 | 288 | 289 | 290 | 291 | 292 | 293 | 294 | 295 | 296 | 297 | 298 | 299 |  | 300 | 301 | 302 | 303 | 304 | 305 | 306 | 307 | 308 | 309 | 310 | 311 |
| Z14 | 312 | 313 | 314 | 315 | 316 | 317 | 318 | 319 | 320 | 321 | 322 | 323 | 324 |  | 325 | 326 | 327 | 328 | 329 | 330 | 331 | 332 | 333 | 334 | 335 |
| Z15 | 336 | 337 | 338 | 339 | 340 | 341 | 342 | 343 | 344 | 345 | 346 | 347 | 348 | 349 |  | 350 | 351 | 352 | 353 | 354 | 355 | 356 | 357 | 358 | 359 |
| Z16 | 360 | 361 | 362 | 363 | 364 | 365 | 366 | 367 | 368 | 369 | 370 | 371 | 372 | 373 | 374 |  | 375 | 376 | 377 | 378 | 379 | 380 | 381 | 382 | 383 |
| Z17 | 384 | 385 | 386 | 387 | 388 | 389 | 390 | 391 | 392 | 393 | 394 | 395 | 396 | 397 | 398 | 399 |  | 400 | 401 | 402 | 403 | 404 | 405 | 406 | 407 |
| Z18 | 408 | 409 | 410 | 411 | 412 | 413 | 414 | 415 | 416 | 417 | 418 | 419 | 420 | 421 | 422 | 423 | 424 |  | 425 | 426 | 427 | 428 | 429 | 430 | 431 |
| Z19 | 432 | 433 | 434 | 435 | 436 | 437 | 438 | 439 | 440 | 441 | 442 | 443 | 444 | 445 | 446 | 447 | 448 | 449 |  | 450 | 451 | 452 | 453 | 454 | 455 |
| Z20 | 456 | 457 | 458 | 459 | 460 | 461 | 462 | 463 | 464 | 465 | 466 | 467 | 468 | 469 | 470 | 471 | 472 | 473 | 474 |  | 475 | 476 | 477 | 478 | 479 |
| Z21 | 480 | 481 | 482 | 483 | 484 | 485 | 486 | 487 | 488 | 489 | 490 | 491 | 492 | 493 | 494 | 495 | 496 | 497 | 498 | 499 |  | 500 | 501 | 502 | 503 |
| Z22 | 504 | 505 | 506 | 507 | 508 | 509 | 510 | 511 | 512 | 513 | 514 | 515 | 516 | 517 | 518 | 519 | 520 | 521 | 522 | 523 | 524 |  | 525 | 526 | 527 |
| Z23 | 528 | 529 | 530 | 531 | 532 | 533 | 534 | 535 | 536 | 537 | 538 | 539 | 540 | 541 | 542 | 543 | 544 | 545 | 546 | 547 | 548 | 549 |  | 550 | 551 |
| Z24 | 552 | 553 | 554 | 555 | 556 | 557 | 558 | 559 | 560 | 561 | 562 | 563 | 564 | 565 | 566 | 567 | 568 | 569 | 570 | 571 | 572 | 573 | 574 |  | 575 |
| Z25 | 576 | 577 | 578 | 579 | 580 | 581 | 582 | 583 | 584 | 585 | 586 | 587 | 588 | 589 | 590 | 591 | 592 | 593 | 594 | 595 | 596 | 597 | 598 | 599 |  |

| Zone | Name |
|---|---|
| Z1 | Jongno-gu |
| Z2 | jung-gu |
| Z3 | yongsan-gu |
| Z4 | Seongdong-gu |
| Z5 | Gwangjin-gu |
| Z6 | Dongdaemun-gu |
| Z7 | Jungnang-gu |
| Z8 | Seongbuk-gu |
| Z9 | Gangbuk-gu |
| Z10 | Dobong-gu |
| Z11 | Nowon-gu |
| Z12 | Eunpyeong-gu |
| Z13 | Seodaemun-gu |
| Z14 | Mapo-gu |
| Z15 | Yangcheon-gu |
| Z16 | Gangseo-gu |
| Z17 | Guro-gu |
| Z18 | Geumcheon-gu |
| Z19 | Yeongdeungpo-gu |
| Z20 | Dongjak-gu |
| Z21 | Gwanak-gu |
| Z22 | Seocho-gu |
| Z23 | Gangnam-gu |
| Z24 | Songpa-gu |
| Z25 | Gangdong-gu |

**Figure 2.** OD matrix, where each OD cell is the location data for each OD movement in the matrix.

3.2.2. Action Model

The simulation included passengers and drivers for each cell in the OD matrix and determined whether or not a match was feasible on the basis of the choice probability of passengers and drivers. Applying the method used by Wu et al. [34], passengers selected a price according to the surge according to a normal distribution. On the other hand, drivers applied choice probability according to a normal distribution with the weight of the centrality analysis outcome in accordance with their destination. Centrality analysis was applied to taxi operation data on the basis of 150 M links provided for Seoul Metropolitan City. The analysis was conducted using data from April to September of 2017 which were processed to generate OD data for each district (gu).

For the choice probability model of drivers and passengers, a surge of 1 (base fare of taxi) was applied by consulting the result of National Bureau of Economic Research(NBER, 2016) study [37]. In addition, the probability followed a cumulative normal distribution using the following equation:

$$\Phi(x) = \int_{-\infty}^{x} N(0,1)(x)dx. \tag{3}$$

The choice probability models of passengers (Equation (4)) and drivers (Equation (5)) according to the surge are expressed below.

$$P\left(accept|P_{offer}\right) = \Phi\left(\frac{P_{offer} - P_{base}}{\sigma}\right) \times W_1, \tag{4}$$

$$P\left(accept|P_{offer}\right) = \Phi\left(\frac{P_{base} - P_{offer}}{\sigma}\right) \times W_2 + \left(\frac{T_rC'_{outD}\left(N_j\right)}{T_rC'_{inD}\left(N_j\right)}\right) \times W_3, \tag{5}$$

where $W_1$ is the weight modifying the choice probability of passengers when surge = 1, $W_2$ is the weight modifying the choice probability of drivers when surge = 1, $T_rC'_{outD}\left(N_j\right)$ is the out-degree centrality value of a specific time round ($T_r$), $T_rC'_{inD}\left(N_j\right)$ is the in-degree centrality value of a specific time round ($T_r$), and $W_3$ is the weight of the equity index allowing differentiation by region.

### 3.2.3. Reward Function

The criteria for reward were different depending on matching. Once a driver was matched with a passenger, the reward was the base fare ($P_{fare}$) for the travel between origin and destination multiplied by the surcharge coefficient ($S_a$). If a driver was not matched with a passenger, a negative reward was applied in line with the waiting time value ($W_t$). The waiting time was set to 8 min considering the fact that the average waiting time during late-night periods when a surcharge is applied is 8.1 min according to a study in Seoul Metropolitan City [38]. The time value ($V_t$) was calculated on the basis of the taxi fare for 1 min in the OD matrix. While it increased profitability through a surcharge, it was also designed to provide a negative reward when unmatched by multiplying the time value for 8 min by the surcharge. The equation for the reward can be expressed as shown in Equation (6). Learning was conducted separately for when a negative value was applied to waiting time and for when it was not. Through a comparison, the study identified the appropriate reward function.

$$r = \begin{cases} matched & P_{fare} \times S_a \\ unmatched & \begin{cases} -W_t \times V_t \times S_a & \rightarrow alt1) \\ 0 & \rightarrow alt2) \end{cases} \end{cases} . \tag{6}$$

### 3.2.4. Q-Learning Algorithm

A total of 20,000 Q-learning reiterations were conducted by taking the index value for each zone into account in the late-night period (Figure 3). In one episode, the goal was to identify the optimal surge for each time slot during 6 h of operation for each OD matrix. The Q-table included states (current state and price) for each time slot and action value by applying the concept of time. The travel time and price required to calculate the reward value (operating profit) for each state referred to a separate OD table. The weight values applied to the preference of drivers were the centrality indices for each time slot and zone. The reward function was created separately depending on the application of the time value.

```
Algorithm parameters: α = 0.1, γ = 0.9, epslion decay = 0.995
Loop for each episode:
    Initialize OD state, as od cell ∈ od matrix
    Loop for each OD state:
        Initialize S, Cumulative time
        Repeat:
    Choose an action(A) from S using policy derived from Q (e.g., Decaying ε-greedy)
        Take action (A), obtain reward R from reward function, and next state S′
Update Q(S, A) ← (1 − α)Q(S, A) + α(R + γ max Q(S′, A)
                                          α
Set S ← S′
        Cumulative Time = Cumulative Time + time
        Until cumulative time less than 6 hour
```

**Figure 3.** Q-learning algorithm pseudo code.

## 4. Analysis Results

### 4.1. The Analysis Result of Areas Expected to Be Marginalized in Ridesharing Services

In order to identify the indices to be applied to reinforcement learning, the taxi operation data in Seoul were categorized into different zones according to 25 districts (gu), and the OD matrix was formed on the basis of the volume of pick-ups and drop-offs using district codes. Consequently, the study analyzed centrality and tried to identify a standardized index for each zone to be applied to reinforcement learning. The origin (the number of pick-ups) was compiled by matching administrative districts through the GIS spatial join function, while the destination (the number of drop-offs) was compiled by using the district(gu) code information for each vehicle departure. Any data including areas surrounding Seoul were excluded.

Using the OD matrix, degree centrality analysis for each zone was conducted, and the average in-degree and out-degree centrality was calculated for weekdays and weekends from 10:00 p.m. to 4:00 a.m. (Figures 4 and 5). According to the analysis, in-degree centrality displayed similar patterns for both weekends and weekdays, whereas the regional deviation was smaller than seen for out-degree centrality. Out-degree centrality was high in the Central Business Districts(CBDs) on weekdays and similarly high on weekends.

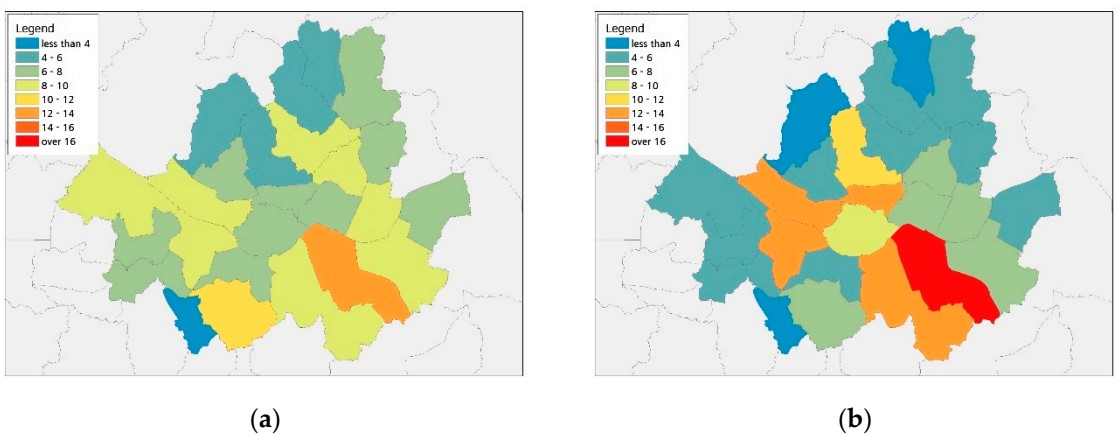

(**a**)　　　　　　　　　　　　　　　　　　　　(**b**)

**Figure 4.** Results of centrality analysis between 10:00 p.m. and 4:00 a.m. on weekdays: (**a**) average in-degree centrality; (**b**) out-degree centrality.

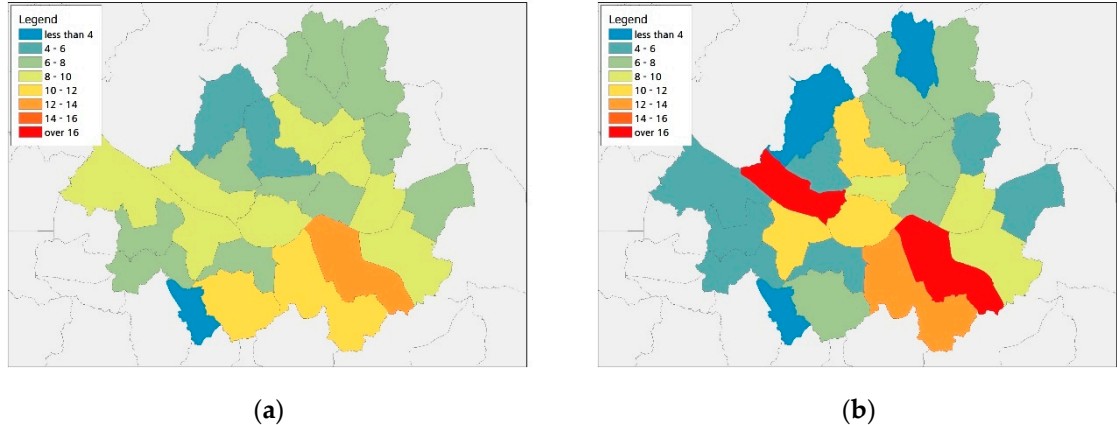

(**a**)  (**b**)

**Figure 5.** Results of centrality analysis between 10:00 p.m. and 4:00 a.m. on weekends: (**a**) average in-degree centrality; (**b**) out-degree centrality.

Areas located on the outskirts of the city had a lower degree centrality. In particular, outskirt areas showed lower out-degree centrality on weekdays. Residential areas on the outskirts of Seoul showed higher in-degree centrality. On the other hand, central and subcentral regions of the city showed higher out-degree centrality (Figure 6). The patterns were similar to those found in previous studies showing the concentration of taxi pick-ups during late-night hours near CBDs [39]. There were large differences in out-degree centrality by zone before and after midnight when most public transportation services were suspended, with the difference growing as the night went on.

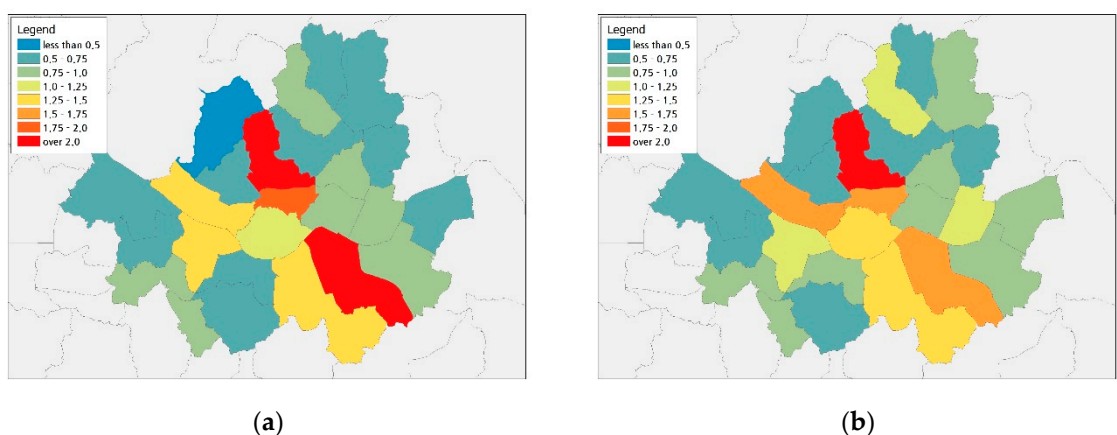

(**a**)  (**b**)

**Figure 6.** Results of out-degree centrality versus in-degree centrality analysis between 10:00 p.m. and 4:00 p.m.: (**a**) weekday; (**b**) weekend.

### 4.2. Results of Reinforcement Learning Simulation

The algorithm presented in Figure 3 was used in the learning of 600 OD cells. When the reward function was defined by a simple matching rate, there was a limitation in comparison among regions as the fare and travel time for each OD were different. Therefore, the simulation implemented a reward value considering travel time and cost. Moreover, analysis was conducted depending on the application of a negative reward value in the unmatched condition. The matching rate for each learning step when a waiting time with a negative reward was applied (alt1) or not (alt2) is depicted in Figure 7.

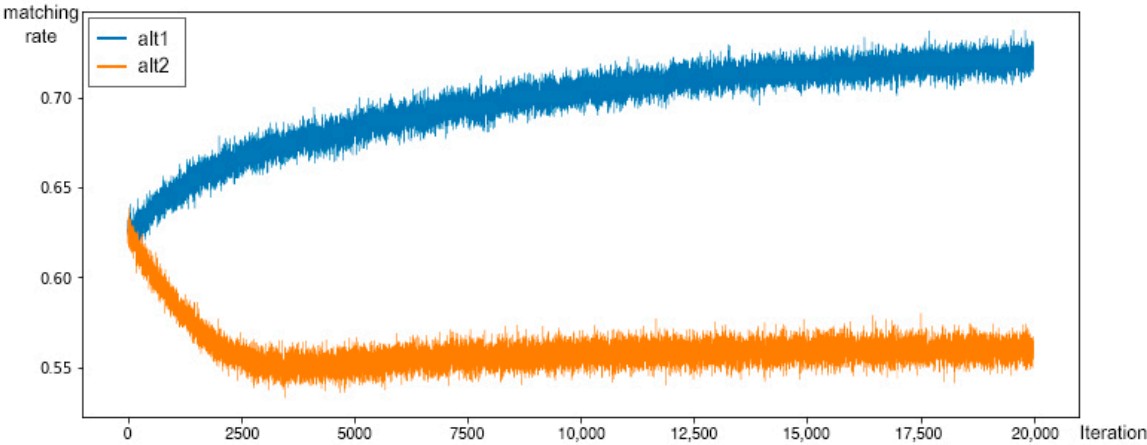

**Figure 7.** Matching rate comparison depending on time value.

When the negative reward was considered, the matching rate increased before converging as learning was reiterated. When it was not considered, the matching rate decreased before converging. According to this result, it can be concluded that the application of a negative reward was necessary when the reward function used the travel cost but not the matching rate. However, the current negative reward for the waiting time used a fixed value from a previous study; thus, for real-world applications, the expected waiting time according to a driver's choice probability should be calculated for each OD.

Using the deduced average surge, when a fare was calculated for each OD, the distribution could be expressed as shown in Figure 8, where the *x*-axis is the travel distance between origin and destination, and the *y*-axis is the price. Then, price levels were compared in terms of the base fare, base fare with late-night surcharge (additional 20% to base fare), late-night surcharge + ride-hail fee (KRW 3000, the highest fee charged by existing platform companies), and surge price (the outcome of learning for each OD). For travel distances shorter than 10 km, the differences among fares were not severe. However, the gap grew wider when the distance increased (see Figure 8a). This is because the late-night surcharge + ride-hail fee (yellow dotted line) was a flat fare regardless of traveling distance, while the surge fare (blue dotted line) was determined using a certain ratio, increasing the absolute price with traveling distance. Table 1 displays the three average prices for different distances, showing the price gap widening as the distance increased.

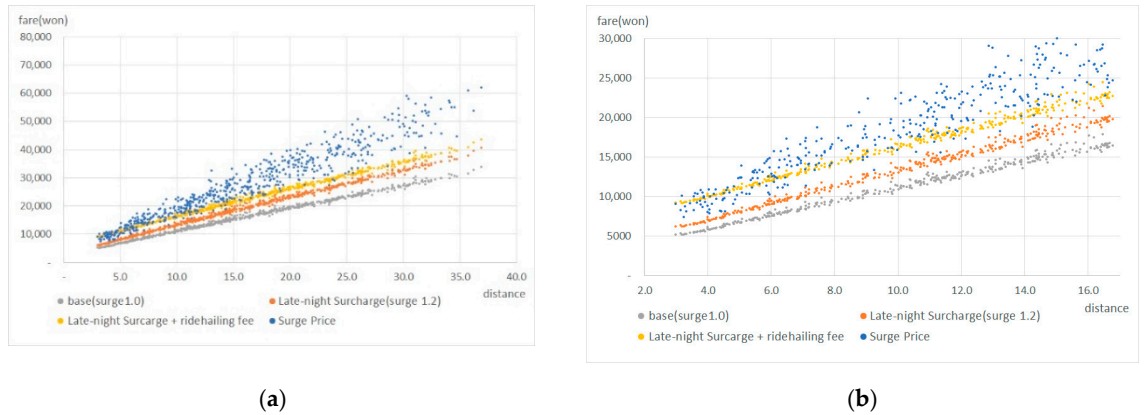

(**a**)                                                                 (**b**)

**Figure 8.** Price distribution for different OD when a surge was applied (using the average surge for different OD): (**a**) total range; (**b**) short distance.

**Table 1.** Traffic volume and average price (KRW) for difference distances.

| Distance | Traffic Volume | Number of OD Zones | Surge Price | Late-Night Surcharge [1] | Late-Night Surcharge + Ride-Hail Fee [1] |
|---|---|---|---|---|---|
| Shorter than 5 km | 28.5% | 46 | 9494.6 | 7017.3 | 10,017.3 |
| 5–10 km | 38.5% | 116 | 14,845.0 | 10,450.8 | 13,450.8 |
| 10–15 km | 18.4% | 146 | 22,264.0 | 15,846.3 | 18,846.3 |
| 15–20 km | 8.5% | 108 | 29,625.4 | 20,863.1 | 23,863.1 |
| 20–25 km | 4.2% | 86 | 36,244.8 | 25,203.1 | 28,203.1 |
| Longer than 25 km | 1.8% | 99 | 45,761.9 | 31,731.4 | 34,731.4 |
| Total | 100.0% | 600 | 27,078.1 | 19,006.2 | 22,006.2 |

[1] The late-night surcharge is surge 1.2 of the base amount and an additional Hail Fee of 3000 KRW (based on the existing platform price).

### 4.3. Changes in Centrality after Surge-Driven Supply Increase

Centrality analysis was carried out after recalculating the traffic volume in accordance with the surge previously identified for each OD matrix and time slot. The traffic volume was recalculated as follows:

$$Nvol_{i,j} = \left(Ovol_{i,j} \times S_{i,j}\right) \times \frac{\sum_{i,j} Ovol_{i,j}}{\sum_{i,j} Ovol_{i,j} \times S_{i,j}}, \tag{7}$$

where $i$, $j$ are the origin and destination, $Nvol_{i,j}$ is the recalculated traffic volume, $Ovol_{i,j}$ is the previous traffic volume, $S_{i,j}$ is the optimal surge for travel (optimal surge for different OD as identified from reinforcement learning), and $\sum_{i,j} Ovol_{i,j}$ is the sum of traffic volume. In centrality analysis, the indicator evaluating the volume of vehicles traveling to a certain zone was the in-degree centrality. When in-degree centrality increases, it can be said that supply is increasing toward that region.

According to the level of improved spatial equity by region, in-degree centrality decreased in the central and subcentral regions of the city, such as Gangnam-gu and Jongno-gu, while the in-degree centrality increased in residential areas on the outskirts of the city, such as Songpa-gu, Gangseo-gu, Dongjak-gu, and Nowon-gu (Figure 9). When this was overlaid on top of the previous hotspot analysis, it was found that the in-degree centrality was drastically improved in districts (gu) located on the outskirts of Seoul Metropolitan City where the number of drop-offs or vacant vehicles was greater compared to the number of pick-ups (Figure 10b). As a result, it can be expected that vehicle supply would be enhanced by as much as 7.5% when applying a higher surge to the region predicted to have a lower choice probability from the perspective of drivers. At the same time, equity in service supply would be improved by reducing the waiting time of passengers in marginalized regions with low taxi demand.

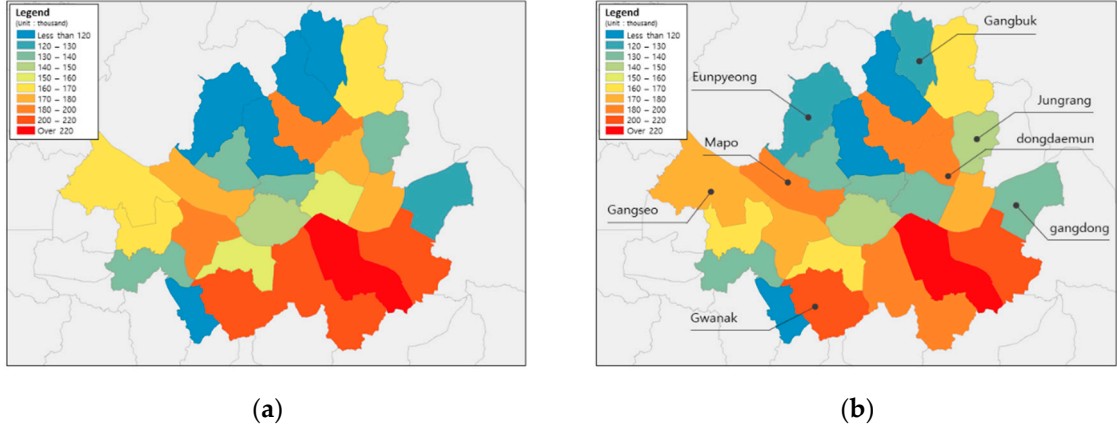

| (a) | (b) |

**Figure 9.** Change in in-degree centrality: (**a**) before; (**b**) after.

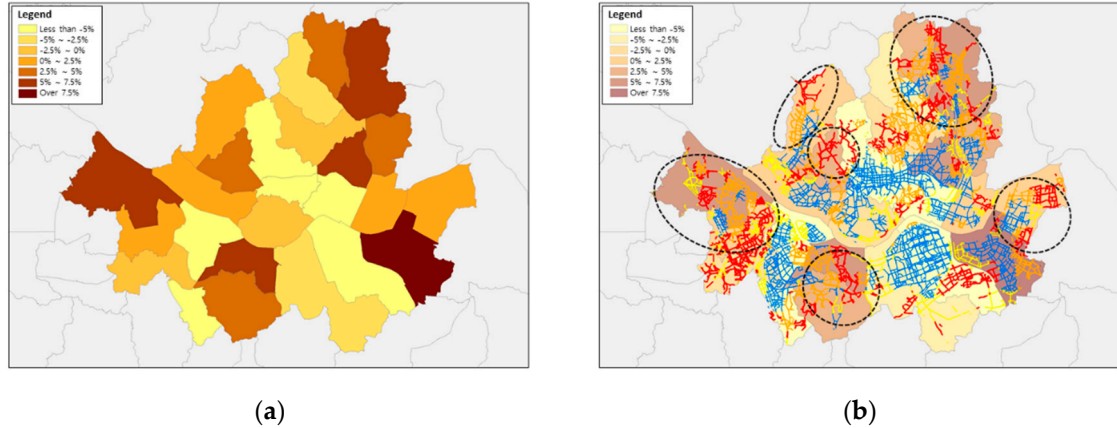

(**a**)                                             (**b**)

**Figure 10.** (**a**) Rate of change in in-degree centrality; (**b**) comparison between in-degree centrality and hotspot analysis.

## 5. Conclusions

As ridesharing (including taxi) services are often run by private companies, profitability is the top priority in operation. This leads to an increase in drivers' refusal to take passengers to areas with low demand where they have difficulties finding subsequent passengers, causing problems such as extended waiting time when hailing a vehicle for passengers bound for these regions. This problem differs depending on time and region. In late-night hours and in suburban regions, the imbalance between supply and demand is especially widened, worsening the problem. In order to address this problem, solutions were proposed in this study through a dynamic pricing strategy using reinforcement learning algorithms.

This study used Seoul city's taxi data to find appropriate fare surge rates for ridesharing services between 10:00 p.m. and 4:00 a.m. In reinforcement learning, the outcome of centrality analysis was applied as the weight affecting drivers' destination choice probability. Moreover, the reward function used during learning was adjusted according to whether or not a passenger waiting time value was applied. Profit was used as the reward value. By applying a negative reward for the passenger's waiting time, a more appropriate surge fare level could be identified. Across the region, the average surge level amounted to 1.6. Regions located on the outskirts of the city in predominantly residential regions such as Gangdong-gu, Dongjak-gu, Eunpyeong-gu, and Gangseo-gu showed a higher surge. On the contrary, central areas, such as Gangnam-gu, Jongno-gu, and Jung-gu, had a lower surge. The findings showed that the supply of ridesharing services in low-demand regions could be increased by as much as 7.5% using surge fares, thereby reducing regional equity problems related to ridesharing services in Seoul to a great extent.

This study conducted a reinforcement learning-based dynamic pricing simulation to respond to the regional equity problem of ridesharing (including taxi) services in Seoul. A novel approach was presented using dynamic pricing as a way to mitigate the spatial equity problem by affecting ridesharing supply, unlike most previous dynamic pricing studies which simply targeted higher profitability. Notably, it was shown that a surge rate change in fares could reduce the indirect refusal of drivers to take passengers to unpreferred areas. With additional real-time ridesharing user data, the Deep Q-Network(DQN) technique can be adopted to conduct a smaller-scale spatial analysis of ridesharing services. Furthermore, with more knowledge on fare sensitivity by user group, the dynamic pricing approach proposed in this study can significantly contribute to resolving the spatial equity problem in mobility services in the future.

**Author Contributions:** Conceptualization, K.Y.H.; data curation, J.S.; investigation, Y.J.C. and M.H.K.; methodology, J.S.; project administration, M.H.K.; software, J.S.; writing—original draft, J.S.; writing—review and editing, K.Y.H. All authors have read and agreed to the published version of the manuscript.

**Funding:** This research was supported by a grant (20TLRP-B148970-03) from the Transportation and Logistics R&D Program funded by the Ministry of Land, Infrastructure, and Transportation of the Korean government.

**Acknowledgments:** This paper is a modification and expansion of Jae In Song's doctoral dissertation.

**Conflicts of Interest:** The authors declare no conflict of interest.

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
