# Peer review of "An Application of Reinforced Learning-Based Dynamic Pricing for Improvement of Ridesharing Platform Service in Seoul"

_electronics, doi:10.3390/electronics9111818_

Round 1
Reviewer 1 Report
The article is very interesting and well written. The Authors applied an interesting research method and obtained valuable conclusions. The article certainly deserves publication.
The aim of the study has been described in the last lines of the Introduction, but I believe that it should be emphasized: the main aim of the study is ...
In the Introduction the research methods and data source on the basis of which the analyzes were performed should be indicated (the first sentence of paragraph 2 could be moved to the introduction).
In the last paragraph of the Introduction, it is worth to present the structure of the article.
At the end of paragraph 2, Authors should highlight their contributions to the literature.
Legends in maps should be enlarged because they are not legible.
There should be a reference to all figures in the text (not only to Figure 4-7).
In Conclusions, the directions of further research in this field should be indicated.
Reviewer 2 Report
Before delving into the contents of the manuscript, I will spend a few words on its style and writing. Overall, I would say that the article is written well enough, meaning that it is easily readable, and the mistakes that I found here and there did not prevent me from understanding its contents. However, maybe the authors would consider submitting their work to a final reading proof, since the small mistakes that are in the text could truly be fixed without much effort. I noticed a general lack of articles throughout the whole text (“Korea saw rapid urbanization” could be “Korea saw a rapid urbanization”, line 30; “greatly improving mobility of citizens” instead of “greatly improving the mobility of citizens”, line 32; “current public infrastructure has its limitations” instead of “the current public infrastructure has its limitation”, line 35-36; etc.), and some imprecisions, such as singular verbs where they should be plural (for instance, lines 34-35: “various policies to facilitate public transportation has been actively promoted.”, where it should be “have been promoted”, since the noun is plural – “various policies”). I think these mistakes are more due to a haste in writing, or maybe a lack of a final check of the English language, rather than to the authors not having a good enough level; so my advice is to fix them, and thus improve the presentation of your work.
Now, moving on to the contents.
Generally speaking, I think this is a good article that, after a few changes, could substantially improve. On the one hand, the introduction is adequate; however, maybe you could include more references, especially in some statements (for instance, line 30. Since the 1970s, with the beginning of industrialization, Korea saw rapid urbanization).
Regarding the methodology and results, some small adjustments should be made in the format. The basis of the figures must be centered below the figure itself, and this is not happening in some cases. However, the tables and formulas are adequately presented. The journal’s template also indicates that a small discussion must be included, where the obtained results are contrasted with other similar researches and new ideas can be produced. In this sense, I recommend exploring other possible factors that affect the problem of equity in spatial mobility, factors other than the price, such as perceived safety. In this sense, I hereby provide a reference that can be useful: Alonso, F., Useche, S. A., Faus, M., & Esteban, C. (2020). Does urban security modulate transportation choices and travel behavior of citizens? A national study in the Dominican Republic. Frontiers in Sustainable Cities, 2, 42.
The conclusion cannot be a mere summary of the results. In this case, you could include how the present research can contribute to the scientific community and/or society. Also, the guidelines of the journal indicate that the “Author contributions” and “conflicts of interest” sections must be included. The reference must be adjusted to the established guidelines as well, since right now they are not following them.
Reviewer 3 Report
The paper addresses significant and interesting problem of rebalancing the fleet and using pricing incentives for both supply (drivers) and demand (travellers).
The method is not introduced properly.
- Equations are using notation which is not introduced and thus the model is not complete.
- How do you simplify taxi operations into your formulas remains unclear.
- The method is mixed with the application (Seul) and assumptions specific to the case-study, it shall be generalisable.
- The results are introducing some of methodology, which shall be introduced before (eq. 4).
- The strategies, that shall come up from the model are hard to understand (eg. How different colours on fig 5 are derived from model assumptions in eq.2 - that link is missing).
- Learning algorithm is not introduced, how do you proceed?
- Is it microscopic, or macroscopic?
- How do you parameterize alpha in Q-table (eq.2)?
- What is the reward for each driver, how is it calculated.
- How does your strategies translate into travellers welfare (waiting time, this can be embedded to table 1).
I think this paper has potential to be published after a major review, when the method is introduced clearly and completely, follow with an illustration of Seoul.
Specific comments:
- Figures are unclear, please provide more labels, captions and explanations
- The centraility measure seems to be inappropriate in my opinion. In network science it identifies nodes crossed by multiple shortest paths, whereas here you do not have paths, but od cells, without paths. So I’d rename it e.g. to ‘balance’ or ‘conservation’ to make it more meaningful.
- Table 1. What do you mean by OD zone? OD is a zone to zone relation. Term ‘sum’ at the end is misleading when you talk about means.
- p5. l 221 please be precise and use notation to introduce the metrics and search space, now it is illegible
- Eq. 3 Fi not introduced in text.
- Eq. 5 illegible, not formatted correctly.
- Fig. 5 slope on a and b shall be the same
- Figures shall be referred from text rather than embedded, in typesetting they will anyhow drift apart, so lets discuss and reference from text.
- Figure 2 - I believe the left and right panels shall be merged to show the difference only (that is what matters, the delta between in and out trips)
- Figure 4. - I do not see relation with 'time value' , do you mean iterations?
- Figure 4. - labels shall be changed to meaningful and in line with text. Now names are misleading.
Reviewer 4 Report
Review of “An Application of Reinforced learning-based Dynamic Pricing for Improvement of ride-sharing Platform Service in Seoul.”
Comments:
In this paper, a study has been conducted on Seoul city’s taxi data to find appropriate surge rates of ridesharing services by region based on reinforcement learning algorithm to resolve the problem during the worst time period. In reinforcement learning, the outcome of centrality analysis has been applied as a weight affecting drivers’ destination choice probability. Moreover, a reward function used in the learning, has been adjusted according to whether the passenger waiting time value is applied or not. In the paper, it has been shown that the supply of ridesharing services in low demand regions can be increased by 7.5% so that regional equity problems related to ridesharing services in Seoul can be reduced to a greater extent. The results are relevant, and I recommend this manuscript for publication in Electronics after some minor revisions.
General comments
- Please write your Keywords in Alphabet.
- Introduction is not coherent and needs to be rewritten.
- Please add the outline of the paper to the end of the introduction. “The rest of this paper is organized as follows; In section 2, …. has been discussed. …. have been presented in sections 3 and 4, respectively. Finally, summary and conclusion are …”
- Please be consistent in the paper. For example: “ridesharing” and “ride-sharing”
